# Omega-7 Mixed Fatty Acid Supplementation Fails to Reduce Serum Inflammatory Biomarkers: A Placebo-Controlled, Double-Blind Randomized Crossover Trial

**DOI:** 10.3390/nu13082801

**Published:** 2021-08-16

**Authors:** Masa Sasagawa, Miranda J. Boclair, Paul S. Amieux

**Affiliations:** 1Bastyr University Research Institute, Kenmore, WA 98028, USA; pamieux@bastyr.edu; 2Department of Naturopathic Medicine, Bastyr University, Kenmore, WA 98028, USA; miranda.boclairl@bastyr.edu

**Keywords:** palmitoleic acid, omega-7 fatty acid, RCT, dietary supplement, biomarkers, placebo effect

## Abstract

We report the effects of mixed omega-7 fatty acid supplementation on changes in serum hsCRP, TNFα, and IL-6 levels and self-reported outcomes in people with non-specific chronic musculoskeletal discomfort. Design: A double-blind, placebo-controlled, 1:1 randomized single crossover trial composed of 688 mg/day palmiteolate for the verum and an equivalent amount of medium-chain triglycerides for the placebo. Method: Data were analyzed in two independent groups and as a crossover group. Results: From 211 screened participants in 2017–2019, 56 were randomized. Six participants dropped out and fifty completers contributed to the statistical analyses. At baseline, none of the investigated biomarkers were significantly correlated to subjectively assessed musculoskeletal discomfort levels. For the two-group analysis (*n* = 26 and *n* = 24), none of the serum biomarkers reached statistical significance; however, a statistically significant placebo effect was found in the subjective outcomes. Conclusion: For the crossover analysis (*n* = 50), three weeks of supplementation with n7FA containing 688 mg per day of palmiteolate did not reduce serum inflammatory biomarkers nor did it improve subjectively measured quality of life (QoL) compared to placebo. Future studies should explore appropriate biomarkers, sufficient power, length of dosing, inclusion criteria for volunteers with higher BMI, and the verification of *cis-*palmiteolate versus *trans-*palmiteolate.

## 1. Introduction

Fatty acids are generally highly permeable to cell and organelle membranes and dietary intake of unsaturated fatty acids such as omega-3, -6, and -9 fatty acids, or various combinations of these, have been studied for their potential health benefits. Postprandial upregulation of genes associated with pro-inflammatory pathways with saturated fatty acid consumption compared to monounsaturated or polyunsaturated fatty acid consumption has also been reported [1]. Essential fatty acids such as eicosapentaenoic acid (EPA) and docosahexaenoic acid (DHA) have been extensively studied for their potential anti-inflammatory effects [2]. Omega-7-rich mixed fatty acid (n7FA) supplements have also been proposed to modulate serum inflammatory biomarkers [3,4]. The principal active constituent of n7FA is palmitoleate, which is a 16-carbon monounsaturated (16:1) fatty acid with a double bond occurring at the 7th carbon atom away from the methyl group. There is no Recommended Daily/Dietary Allowance (RDA) for palmitoleate, which is not an essential fatty acid, meaning the body can produce it de novo [5].

A double-blind, placebo-controlled, 1:1 randomized single crossover (3 weeks × 2) trial of an omega-7 fatty acid (n7FA) supplement containing palmitoleate was conducted from 2017–2019. The investigational supplement was provided by Barlean’s Organic Oils, LLC (Ferndale, WA, USA), which was also a sponsor of this trial together with Bastyr University Research Institute. The aim of this study was to replicate the findings reported by Bernstein et al. on high-sensitivity C-reactive protein (hsCRP), and to correlate this with two other serum inflammatory biomarkers: tumor necrosis factor-alpha (TNFα) and interleukin-6 (IL-6). By enrolling a sufficient number of eligible volunteers to reach 50 participants who would provide data on three time points, the data from Bernstein et al. suggested that statistical significance would be achieved.

## 2. Materials and Methods

### 2.1. Administrative Considerations

The registration of this trial to the clinicaltrial.gov site on 13 September 2018 was delayed due to a contractual disagreement that occurred during the recruitment prior to data analysis (https://clinicaltrials.gov/ct2/show/NCT03669575, accessed 13 September 2018). The trial was conducted in accordance with the International Council of Harmonisation of Technical Requirements for Pharmaceuticals for Human Use (ICH) and Good Clinical Practice (GCP). The protocol was approved by the Bastyr University Institutional Review Board (IRB#16-1575). Informed consent was obtained from all study participants.

Due to the sponsor company being a longtime supporter of the university, the Office of Research Integrity (ORI) required that the research team maintain a strict and conscientious boundary between the sponsor organization and the research team. Our ORI is the administrative office for the IRB and oversees the scientific and ethical integrity of research, including managing any potential conflict of interest (COI). The a priori protocol and double-blinding were strictly maintained, and all acquired data per protocol were used for the analysis. All analyses, including tallying adverse events, dropouts, and crossover analyses were performed while a statistician was given the A or B designations only. Outliers were detected by the Grubb’s Extreme Studentized Deviate (ESD) test. The code was broken in a group meeting after all analyses were completed. The trial took place at the Clinical Research Center at Bastyr University in Kenmore, Washington.

### 2.2. Double Blinding and Randomization

The investigational products were transported to the trial site with dosing and storage instructions. Each bottle was stamped with a unique lot number, which distinguished a placebo or verum bottle identifiable by a single designated researcher. All staff including the PI were blinded to the bottle identity. Fifty-six bottles of verum and 56 bottles of placebo (112 bottles of study product) were prepared. Fourteen blocks of four (2 verum and 2 placebo) were randomized as A or B, assuring an equal number of participants assigned to both groups for every four subjects enrolled [6]. Each participant received one of the verum and one of the placebo bottles based on the sequence defined by the A or B designation.

### 2.3. Verum and Placebo

Per the certificate of analysis, the verum capsules contained 413 mg omega-7 Fish Oil Blend (Barlean’s) derived from Alaskan Pollock (Gadus Chalcogrammus) consisting of 227 mg palmitoleate per capsule. The rancidity of the oil was measured by the level of peroxide and p-Anisidine, indicated as 1.16 mEq/kg and 3.91 mEq/kg, respectively. Peroxide values (PVs) under 10–20 mEq/kg of fish oil are considered very fresh and unlikely to exhibit rancid smells [7]. Hung and Slinger (1980) have indicated that for salmon oil, a PV of 26 mEq/kg is ‘moderately oxidized’, a PV of 120 mEq/kg is ‘highly oxidized,’ and a PV of 314 mEq/kg is ‘extremely oxidized’ [8]. Placebo capsules contained the equivalent quantity of fatty acid in the form of medium-chain triglycerides (MCT) consisting of 65% caprylic acid (8:0), 45% capric acid (10:0), and <2% lauric acids (12:0) per the company’s information. The investigational products were stored at room temperature in a locked closet.

### 2.4. Biomolecule Measurements (Plasma Total Fatty Acid/Cytokines/hsCRP)

The total fatty acid composition and amounts thereof in placebo and verum capsules and participant plasma samples were determined via lipid extraction based on the method of Bligh and Dwyer in the presence of known amounts of added tridecanoin (C13:0 triglyceride) and methyl tricosanoate (C23:0 Methyl Ester) in order to determine total lipid/total fatty acids [9]. An aliquot of the total lipid extract (lower phase) was taken for quantitation of the fatty acids following transmethylation as described by Morrison and Smith [10]. The fatty acid methyl esters were prepared using boron trichloride in methanol and heating the methylation tubes in a heating block at 95 °C for 50 min. The fatty acid methyl esters were then analyzed on an Agilent 7890B gas-liquid chromatographer with a 60-m DB-23 capillary column (0.32 mm internal diameter) using a standard mixture(s) for both qualitative and quantitative analysis along with the known fatty acid components for retention time verification. Internal standards were obtained from NuChek Prep, Elysian, MN, USA). Quality control standards were purchased from Millipore-Sigma Canada (Oakville, ON, Canada). Recovery was determined by comparison of peak area between internal standards (90–110%). Plasma specimens were sent to Lipid Analytical Laboratories in Ontario, Canada.

Cytokines were analyzed by the Bio-Plex Suspension Array System and Milliplex platform [11]. Plasma specimens were sent to the Eve Technologies Corporation, Ontario, Canada.

Serum specimens were sent to Labcorp, Inc., for hsCRP and lipid panel analysis.

### 2.5. Statistical Calculations

Statistical power was calculated based on the report of *cis-*palmitoleate dosing of 220.5 mg/day for 30 days, which lowered the serum hsCRP by −1.9 mg/L (95%CI: −2.3 to −1.4) relative to a control, which was 1000 mg/day dosing with medium-chain triglycerides [12]. Based on this power calculation, four subjects per group were needed to reach a statistical power of 80% at an α-level of 0.05. Our study proposed to use 454 mg/day for 21 days (per the company’s certificate of analysis), and to explore changes in two other anti-inflammatory cytokines, TNFα and IL-6, by increasing the number of subjects to 56 (15% drop out estimated).

Using a crossover design, 28 participants received the verum first and another group of 28 participants received the placebo first, and this data was then analyzed by a mixed pre-post t-test. Blood was collected from the antecubital vein. In addition to the objective outcomes, three subjective measurements were selected from the questionnaire-bank of the REDCap database system [13], which were: (1) fatigue, (2) pain interference, and (3) physical function domains from the Patient-Reported Outcomes Measurement Information System (PROMIS^®^) [14]. These instruments were filled out at the baseline, the first follow-up, and the last visit.

### 2.6. Crossover Trial Design, Procedures, and Sequence of Events

Volunteers with uncomplicated chronic musculoskeletal discomfort or a baseline serum hsCRP >1.0 mg/L were recruited from 2017–2019. Participants took two capsules per day from the first bottle for three weeks followed by the same dosage from the second bottle without washout. Participants were instructed to maintain their current diet and lifestyle for the six-week period. The trial was characterized by the following: (1) receiving voluntary contact, (2) In-person Informed Consent Form was signed, (3) the baseline assessment, (4) no run-in re-assessment, (5) taking two gel capsules a day from the first assigned bottle for three weeks, (6) returning for the first follow up assessment, (7) no washout period, (8) taking two gel capsules a day from the second assigned bottle for three weeks, (9) returning for the last assessment. The crossover design aims to eliminate between-subject variability; however, the level of sensitivity and the power of statistical calculations are determined by three different assumptions between the verum and placebo responses: (1) having a positive correlation, (2) having no correlation, or (3) having a negative correlation [15]. A positive correlation between the verum and the placebo means that if you have a large placebo effect, your verum effect will also be greater. A negative correlation between the verum and the placebo means that a high placebo responder will have a smaller effect on the verum. In this study, no correlation was assumed.

## 3. Results

Of the 211 subjects who consented and screened, 56 subjects were randomized. Five subjects dropped out during the verum dosing and one subject dropped out during the placebo dosing, which resulted in 50 subjects, as shown in Figure 1. At the baseline, twelve subjects (24%) were using some sort of dietary supplement such as a multivitamin, fish oil, turmeric; alone or in combination with other pharmaceuticals. Non-steroidal anti-inflammatory drug use was 16%; prescriptions of muscle relaxants or non-opioid and non-barbiturate pain medications was 12%; and 8% of enrollees were using other pharmaceuticals such as anti-hypertensives or antidepressants. Overall, the participants were generally healthy adults with chronic musculoskeletal discomfort; however, Group B showed healthier biomarker characteristics compared to Group A.

At the baseline, none of the plasma biomarkers, including hsCRP, was statistically significantly correlated with the musculoskeletal discomfort assessment. Demographic information is summarized in Table 1. Although some of the characteristics show statistically significant differences between the two groups, the crossover design allowed both groups to contribute to the aggregate statistics for placebo and verum.

### 3.1. Analysis of Compliance and Adverse Events

The level of compliance was assessed by counting the remaining pills in the returned bottles. The following three numbers indicate: >75% compliance, <75% compliance, and missing bottles: Verum group: 41, 5, 4; placebo group: 35, 8, 7. The number of compliant subjects for both bottle groups (>75%) was 32. New symptoms reported did not require urgent care and the symptoms were resolved within three days. The adverse events reported for verum were: gasses (1), loose stool (1), GI symptoms (1), digestive issues (1), drowsiness (1), constipation (1); and for placebo: burping (1), loose stool (1), gastritis (1), low back tweak (1), drowsiness (1), and headache (1).

### 3.2. Two Independent Group Analysis

For the non-crossover analysis, Figure 2 indicates the baseline, followup_1, and followup_2 measurements of Group A (*n* = 26) and B (*n* = 24) for three objective and three subjective measurements. Three biomarkers demonstrated no pre-post significance for verum and placebo, whereas subjective measurement showed statistically significant reductions in Fatigue and Pain Interference measurements between the baseline and followup_1 for the placebo only; however, this reduction did not continue to the third measurement (followup_2).

### 3.3. One Crossover Group Analysis

The placebo data were collected from 26 data points in Group A and 23 data points in Group B, and the verum data were collected from 24 data points in Group A and 26 data points in Group B (see Figure 1). The aggregate average of the difference within each subject (*n* = 50) was then compared between the placebo and verum by multiple *t*-tests. One outlier in the placebo data was identified by the Grubb’s ESD test and excluded in series from the aggregate calculations. Using the Holm–Sidak multiple-comparison adjustment, none of the factors was statistically significantly different between the verum and placebo (Table 2 and Figure 3).

### 3.4. Fatty Acid Analysis

The placebo capsule, verum capsule, and plasma, were analyzed for 13 fatty acids. The frozen plasma specimens of seven subjects who were higher than 90% compliant by pill count were selected for fatty acid analysis to investigate whether or not an elevated quantity of n7FA was detectable when verum was dosed compared to placebo (Table 3).

## 4. Discussion

The key points are:Three weeks of a 688 mg/day palmitoleate mixed fatty acids dietary supplement in a small, randomized, single crossover trial did not produce a statistically detectable change compared to an MCT placebo dietary supplement.When each time point was examined, improved subjective PROMIS^®^ measures of fatigue and pain interference were observed between the baseline and first follow-up assessment for the placebo only by Fisher’s LSD adjustment (*p* < 0.03, pre-post paired F(2,75) = 2.51, two-tailed).

The concept for this project originated from a small palmiteolate dietary supplement clinical trial published in 2014 [12]; however, after completion of this study, we found the original article had been retracted with the following statement: “this article presents serious concerns about Food and Drug Administration guidance and about data or statistical interpretation. The statistical accuracy claimed in the published article is not consistent with known variability of lipoprotein cholesterol and triglyceride levels” [12]. Although this explanation does not address the validity of the n7FA results, our results did not support the claim made by Bernstein et al. (2014). Researchers investigating the anti-inflammatory effects of n7FA supplements have measured cytokines in vitro such as TNFα, IL-6, IL-8, and monocyte chemoattractant protein-1, and have compared them to inflammatory fatty acids such as oleic and palmitic acids [3], or proposed the modulation of SCD1 [16]. Our study showed a decreased trend for the cytokines TNFα and IL-6 with MCT and an increased trend with n7FA. It is a well-described phenomenon that most natural product clinical trials have failed to show statistical significance beyond placebo [4]. CAM clinical trials, including vitamins, minerals, and other dietary supplements are likely to have low effect sizes because the underlying purpose of using CAM is most often to assist the inherent healing process of the individual and this may be easily confounded by the placebo effect.

Because de novo synthesis of n7FA involves stearoyl-CoA desaturase 1 (SCD1) acting on palmitate (16:0) to produce 16:1n7FA17, the desaturation ratio (16:1n7/16:0) could be influenced by dietary intake of n7FA. However, the desaturation ratio or other free fatty acid concentrations alone did not distinguish plasma from baseline, verum, or placebo. For instance, we expected an elevation of medium-chain fatty acids in the plasma of placebo-dosed participants; however, fluctuation in the values was negligible. We were unable to find a particular pattern in the fatty acid profile by dosing, and the timing of the last dose was not documented. Because the visiting appointment was usually in the morning, the last dose was likely the night before or 10–12 h prior to the blood draw. Two additional findings were: (1) contrary to the certificate of analysis provided (227 mg/capsule 16:1n7FA), our third-party independent analysis found the quantity of 16:1n7FA to be 344 mg per capsule, and (2) the verum also contained 199 mg per capsule of palmitate. Therefore, the dosage used in our verum would be 688 mg of 16:1n7FA and 398 mg of palmitate per day (see the Materials and Methods Section 2.4 for analysis details).

As for the limitations of this study, different n7FA products may account for different outcomes. The chloroform-methanol extraction method for fatty acid quantification was reported to be less than solvent-free triglyceride determination of membrane-bound fatty acids [17]. Furthermore, the use of MCT as the placebo control, although not directly affecting the crossover design, clearly was a questionable choice given the fact that MCT has demonstrated anti-inflammatory activity [18,19]. The original choice of the placebo was to maintain the same caloric value as the verum without a plan to use an active placebo. The crossover design aims to eliminate between-subject variability; however, incomplete washout can contaminate the effects of placebo and verum. Participants were mostly female, but gender effects cannot be generalized due to the small study size. The plasma fatty acid analysis did not contribute useful information to discern group assignment or bioavailability data.

This study strongly suggests that researchers should pay attention to fatty acids that are absorbed into the body quickly. Neither the orally taken MCT fatty acids nor the n7FA appeared in the plasma at detectable levels after 12 h as measured by plasma total lipid/total fatty acid content. Bradbury et al. (2011) suggested that the 16:1 fatty acid detected would be 16:1n7 (with no 16:1n9); and for human serum/plasma lipids, the 16:1n9 would be undetectable or only trace amounts detected [20]. Bioavailability data on the essential fatty acid (n3FA) showed a maximum absorption peak at 7–10 h after oral ingestion [21]. In the case of non-essential FAs, biofeedback mechanisms may play a role in controlling the body’s FA concentration even within three weeks. A comparison study of long-term low dose versus short-term high dose non-essential FAs may provide additional information. For bioavailability detection, the analysis of red blood cell membranes would be advised for future studies.

When designing our study, pure 16:1n7FA was both cost-prohibitive and not safe for use as a control because the LD_50_ has not been established [22]. Dietary supplement companies regularly refer to their product as ‘cis-palmitoleate,’ perhaps because partially hydrogenated fatty acids or processed trans-isoforms have been shown to be harmful [23]. Traditionally, researchers have assumed that the endogenously produced or naturally occurring palmitoleate is the cis-isoform; however, trans-palmitoleate can be naturally produced from the enzymatic shortening of dietary vaccenic acid in vivo [24] as well as in dairy products, and these forms of trans-palmiteolate act like lipokine hormones [25,26]. A healthy French-Canadian population study found that high fat dairy intake was positively correlated with plasma trans-palmitoleate levels (r = 0.15; *p* = 0.03), and total dairy consumption was positively correlated with plasma CRP (r = 0.15; *p* = 0.03) [27]. Other large cross-sectional dietary studies report the association of plasma trans-palmitoleate with: (1) high intake of whole-fat dairy products [28]; (2) inflammatory biomarkers in an elderly Swedish population [29]; and (3) other inflammatory markers such as hsCRP, RANTES, IL-1Ra, interferon-gamma, IL-10, and PDGF-bb [30]. Therefore, verification of the cis- versus the trans- form of palmitoleate used in any future clinical trials on n7FA may prove essential. Future investigators should also consider conducting a proof-of-principle intervention to demonstrate whether omega-7 supplementation results in greater plasma or lipid membrane n7FA and the intervention should also test for other fatty acid metabolites of n7FAs.

From the viewpoint of clinicians, this study may be criticized for a lack of clinical insight because subjects were selected by voluntary participation rather than active selection by clinicians who might identify patients who would benefit from supplementation with n7FA in order to alleviate musculoskeletal discomfort. For example, Hsu et al. (2019) also reported that the level of serum hsCRP was not statistically different for chronic kidney disease patients with or without musculoskeletal pain [31]. The authors acknowledge this limitation; however, volunteerism and transparency are the nature of human subjects research. As n7FA supplements are readily available and circulating without prescription or FDA approval, typical consumers are not clinically trained but self-dosing. Thus, this study focused on the typical consumer population rather than a clinically handpicked population.

## 5. Conclusions

In our sub-analysis, subjects with higher hsCRP levels at the baseline showed no improvement with 3 weeks of supplementation with 688 mg of palmiteolate, nor did they show an improvement in subjectively assessed musculoskeletal discomfort beyond placebo (data not shown). Future studies should consider: (1) identifying individuals with elevated biomarkers and comorbidities such as high BMI; (2) the length of dosing with n7FA; and (3) prior determination of what percent of the palmiteolate fatty acid dietary supplement is in the cis-isoform. Our study suggests that for the general population with a range of hsCRP levels and subjectively assessed musculoskeletal discomfort, there is no benefit to n7FA supplementation.

## Figures and Tables

**Figure 1 nutrients-13-02801-f001:**
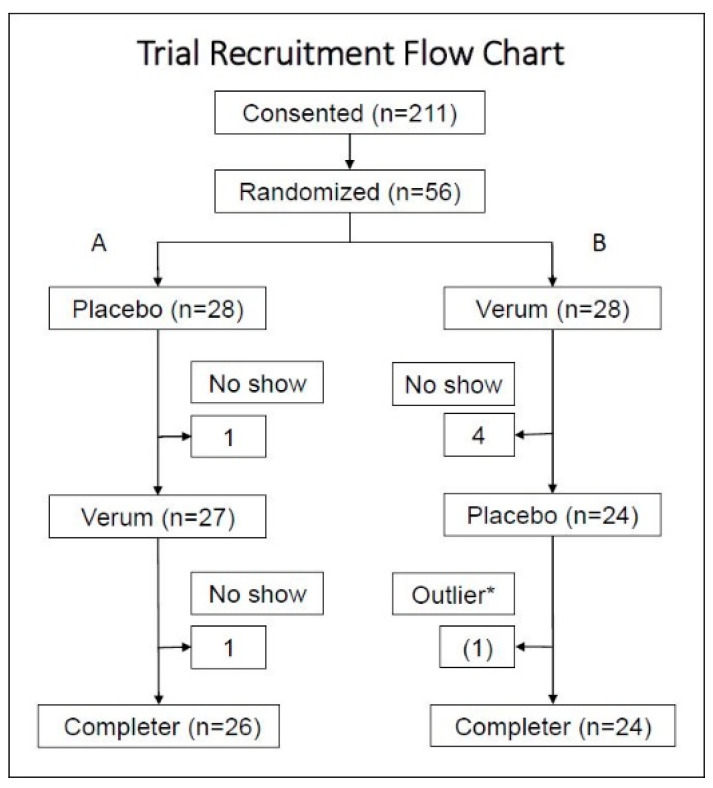
After screening and consenting, subjects were randomized to either Group A (placebo followed by verum) or Group B (verum followed by placebo). With the single crossover design, subjects who participated in both verum and placebo measurements were included in the analysis per protocol. * One hsCRP data point was detected as an outlier by the Grubb’s HSD at the alpha = 0.05. This subject reported a flu episode on the date when the outlier was detected. This subject was not excluded; however, for aggregate statistics, the data points (follow-up 2) of this subject were excluded in series.

**Figure 2 nutrients-13-02801-f002:**
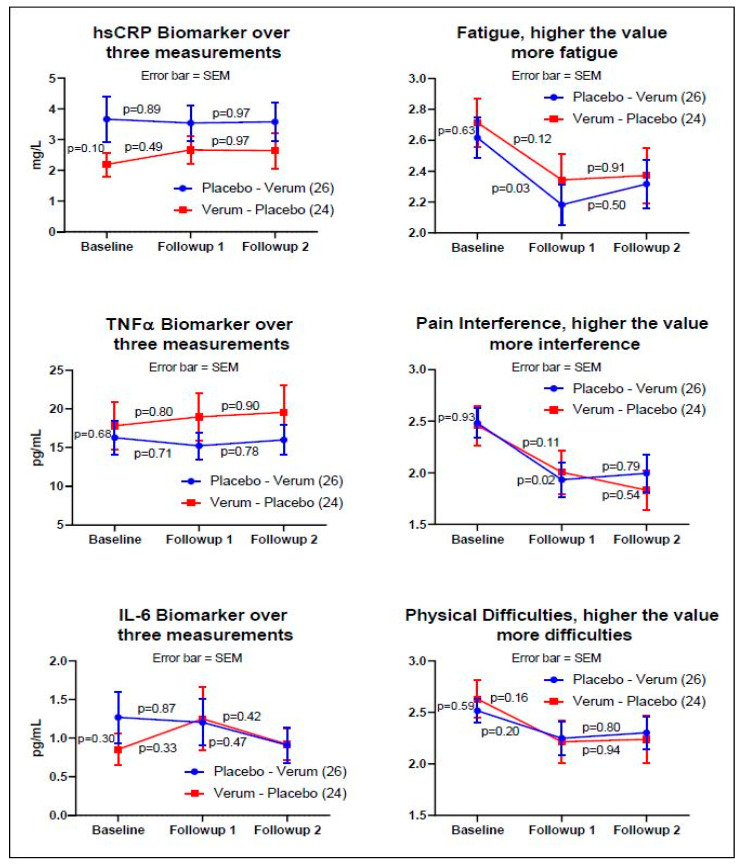
The graphs on the left side are the biomarkers: hsCRP, TNFα, and IL-6. The graphs on the right side are the subjective assessments: PROMIS^®^ fatigue, pain interference, and physical functioning. The y-axis values on the right side of the subjective assessments are the averages of the raw scores from each validated PROMIS^®^ measure (fatigue, pain interference, physical function) on a 5-point Likert scale. The blue line indicates Group A, and the red line indicates Group B. The *p*-values indicate pre-post paired-*t* tests adjusted by the Fisher’s LSD. Subjective measures demonstrated a reduction in three PROMIS^®^ measures at the time of entry into the study to the first follow up visit; however, the reduction did not continue for either the placebo or the verum dosing. Statistical significance was shown for the subjective assessment of fatigue and pain interference on the placebo arm only (*p* < 0.03, two-tail test, Fisher’s LSD).

**Figure 3 nutrients-13-02801-f003:**
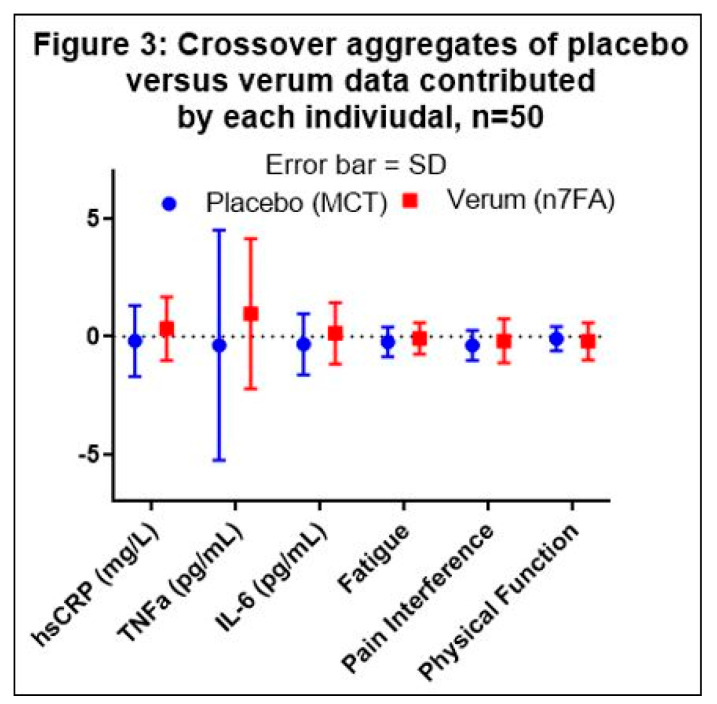
Graphical presentation of Table 2. Medium Chain Triglycerides (MCT) demonstrate a consistent anti-inflammatory effect compared to omega-7 fatty acid (n7FA). The error bars represent the standard deviation (SD). None of the comparisons between placebo and verum were statistically significant.

**Table 1 nutrients-13-02801-t001:** General Health Characteristics of Group A and Group B Subjects.

Title 1	Group A (*n* = 26)	Group B (*n* = 26)	Multiple *t*-Test
Age	55.5 ± 13.2	56.3 ± 14.4	0.84
Female	20 (6 males)	21 (3 males)	n/a
BMI (kg/m^2^)	30.4 ± 7.6	28.2 ± 7.0	0.29
Cannabis use	5	4	n/a
Type of diet reported	Ketogenic/Low carb (4)	Ketogenic/Low carb (5)	n/a
	Vegetable-based (2)	Vegetable-based (5)	
	Gluten free (3)	Gluten free (1)	
	Low fat (1)	Low fat (2)	
	Other healthy (5)	Other healthy (3)	
	Std American (11)	Std American (8)	
Exercise Frequencies			
Never	2	2	n/a
1–2 × per month	1	2	n/a
Once a week	4	5	n/a
2–3 × per week	7	5	n/a
4–5 × per week	10	7	n/a
>5 × per week	2	3	n/a
Biological Measurements (mean ± standard deviation)			
hsCRP (mg/L)	3.7 ± 3.8	2.2 ± 1.9	0.10
Total Cholesterol	222.0 ± 37.9	208.3 ± 45.2	0.25
HDL (mg/dL)	54.7 ± 15.0	65.5 ± 20.4	0.04 *
LDL (mg/dL)	130.3 ± 35.3	115.1 ± 43.5	0.18
VLDL (mg/dL)	37.0 ± 17.2	27.8 ± 16.2	0.06
TG(mg/dL)	184.8 ± 87.7	139.3 ± 81.5	0.01 *
TNF (pg/mL)	16.3 ± 11.0	17.8 ± 15.2	0.68
IL-6 (pg/mL)	1.3 ± 1.7	0.9 ± 1.0	0.30

* indicates significance at the alpha level *p* ≤ 0.05. In general, subjects in Group B appeared to display better lipid profiles; however, this should not be a critical randomization error for a cross-over design. The type of diet and exercise frequencies reported showed that the participants were fairly health conscious. For instance, the ketogenic diet was a trendy diet in 2017, and 14% were vegetarian or vegetable-based diets. More than 50% of participants exercise at least every other day. Multiple t-tests were adjusted by the Holms–Sidak method. Abbreviations: Body Mass Index (BMI); high sensitivity C-reactive Protein (hsCRP); high-density lipoprotein (HDL); low-density lipoprotein (LDL); very-low-density lipoprotein (VLDL); triglyceride (TG); tumor necrosis factor (TNF); interleukin-6 (IL-6).

**Table 2 nutrients-13-02801-t002:** Crossover Group Analysis.

	M ± SD	N	Difference ± SEM	Df	Holm-Sidak Method Adjusted *p*-Value
hsCRP_p	−0.07 ± 1.70	49			
hsCRP_v	0.27 ± 1.51	50	−0.31 ± 0.32	97	0.80
TNFα_p	−0.30 ± 4.85	49			
TNFα_v	0.94 ± 3.16	50	−1.24 ± 0.82	97	0.58
IL6_p	−0.21 ± 1.50	49			
IL6_v	0.04 ± 1.45	50	−0.25 ± 0.30	97	0.80
Fatigue_p	−0.23 ± 0.62	49			
Fatigue_v	−0.11 ± 0.68	50	−0.13 ± 0.13	97	0.80
PainInt_p	−0.35 ± 0.59	49			
PainInt_v	−0.18 ± 0.82	50	−0.17 ± 0.14	97	0.76
PhyFunc_p	−0.11 ± 0.49	49			
PhyFunc_v	−0.17 ± 0.73	50	0.06 ± 0.12	97	0.80

Fifty data points (n = 50) were provided from 26 subjects who were assigned to taking the placebo-verum sequence and 24 subjects who were assigned to the verum-placebo sequence. A mixed *t*-test or split-plot analysis showed no significant difference between the verum and placebo measurements in both biomarkers and subjective assessments. One data point in the placebo measurement was found to be an outlier by the Grubb’s ESD test at an alpha level of 0.05. The multiple *t*-tests were adjusted by the Holm–Sidak adjustment. Abbreviations: mean plus or minus standard deviation (M ± SD); number of subjects (N); difference plus or minus standard error of the mean (Difference ± SEM); Df = degrees of freedom; hsCRP = high sensitivity C-reactive Protein; TNFα = tumor necrosis factor-alpha; IL-6 = interleukin-6; PainInt = pain interference; PhyFunc = physical functioning; “p” or “v” after abbreviations indicate p = placebo or. v = verum, respectively.

**Table 3 nutrients-13-02801-t003:** Analysis of Fatty Acids in Placebo Capsules, Verum Capsules and Plasma.

	Placebo Capsule (mg/g)	Verum Capsule (mg/g)	Baseline Plasma (µg/mL)	Placebo Plasma (µg/mL)	Verum Plasma (µg/mL)
C8:0	526.58 ± 0.12	9.74 ± 1.13	0.04 ± 0.10	0.09 ± 0.24	0.00 ± 0.00
C10:0	359.20 ± 0.76	6.15 ± 0.78	1.05 ± 0.73	1.19 ± 1.28	1.02 ± 0.92
C12:0	0.44 ± 0.00	0.04 ± 0.01	0.86 ± 1.89	0.61 ± 1.25	0.12 ± 0.08
C16:0	0.23 ± 0.03	261.65 ± 0.91	1065.96 ± 282.45	995.13 ± 210.52	939.40 ± 308.84
C16:1n7	2.14 ± 0.10	452.16 ± 1.72	84.68 ± 41.96	71.40 ± 37.19	78.29 ± 46.55
C20:5n3	0.10 ± 0.07	1.71 ± 0.23	29.81 ± 14.82	31.52 ± 18.79	27.04 ± 10.08
C22:6n3	0.21 ± 0.07	4.92 ± 0.17	71.73 ± 53.74	78.09 ± 46.40	61.11 ± 23.85
Total	891.28 ± 0.01	761.92 ± 2.34	n/a	n/a	n/a

The major constituents of placebo capsules were capryliate (8:0), capriate (10:0), and laurate (12:0); whereas the major constituents of the verum capsules were palmitate (16:0), palmitoleate (16:1), eicosapentaenoic acid (EPA, 20:5n3), and docosahexaenoic acid (DHA, 22:6n3). Statistical analysis (ANOVA) demonstrated that none of the fatty acids listed differed in the plasma at the baseline, nor during placebo and verum dosing. In other words, although subjects were taking a palmitoleate-rich capsule during the verum dosing, the plasma fatty acid profile did not show a statistical difference. (*n* = 7 who were >95% compliant by pill count).

## Data Availability

Seven years from the date of publication at Bastyr University. De-identified data; however, a Data Use Agreement (DUA) should be signed before the transfer of data.

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
