# Peer review of "Omega-7 Mixed Fatty Acid Supplementation Fails to Reduce Serum Inflammatory Biomarkers: A Placebo-Controlled, Double-Blind Randomized Crossover Trial"

_nutrients, 2021, doi:10.3390/nu13082801_

Round 1

Reviewer 1 Report

Sasagawa et al studied the effect of omega 7 fatty acid supplementation and its effect on serum biomarkers of inflammation with a small placebo-controlled double-blind randomized crossover trial. The trial was intended to replicate the finding of a previous study by Bernstein et al.

Major comments:

The investigators assessed compliance by counting pills in the returned bottles and found about 62% of participants had >75% compliance. They also measured the levels of fatty acids before and after supplementation in some of their most compliant participants and found that there was essentially no change in fatty acid compositions in those participants.

The reason for this befuddling and misleading finding is due to the fact that fatty acid measurement was done in the free fatty acid fraction. Had the authors measured fatty acid composition in either the serum phospholipid fraction or RBC, useful information would have been obtained on the extent of changes in palmitoleic acid after supplementation. In turn, such results may shed light on the adequacy of supplementation, extent of compliance vis a vis changes in biomarkers of inflammation and greatly strengthen the impact of the investigation.

Assay for serum phospholipid fraction fatty acids composition are available in many research and (some) clinical laboratories. Thus I strongly recommend that the authors perform this relatively easy assay on a subset of the participants (choosing from both highly and moderately compliant participants per pill count) as adding this information would greatly strengthen the impact of the findings in this investigation.

Minor comments

  1. Two study designs might have affected the results: the use of medium chain triglyceride capsule in the placebo group (MCTs are known to lower markers of inflammation PMID 25458829, 32643946) and the lack of a washout period. The authors should further emphasized these limitations in their discussion.
  2. Explain the process of the final selection of 56 participants from 211 consented participants. What are the exclusion/inclusion criteria used in the selection of 56 from 211.
  3. It might be helpful to include a short and clear explanation on how the measurement of 16:1n7FA in capsule led to the conclusion that participant were actually taking 688mg of 16:1n7FA and 398mg of palmitate per day.

Reviewer 2 Report

Comme

Review: Omega-7 mixed fatty acid supplementation fails to reduce serum inflammatory biomarkers: A placebo-controlled, double-blind randomized crossover trial

Summary: The investigators examined the effects of omega-7 supplementation on plasma inflammatory markers and subjective measures of quality of life over the course of a three weeks in a placebo controlled double blind cross-over trial (3 weeks x2 for placebo). N-7 supplementation was not found to influence the outcomes of interest, and borderline findings for two subjective measure of quality of life were reported for the placebo at one time point after baseline.

Comments

Major

One of the most substantial issues in this trial was the failure of the n-7 treatment to alter plasma n-7 levels over the 3-week time frame. This would appear a potential fatal flaw as the investigators have no evidence that the intervention resulted in n-7 incorporation into plasma phospholipid or cholesterol ester fractions or cell membranes—what likely amounts to the necessary component for producing downstream effects, unless there is some unmeasured downstream metabolite that is otherwise affected. Given this validation problem, it is difficult to see the value of the trial’s null findings apart from providing information to subsequent investigators that omega-7 supplementation: 1) may not alter an individual's fatty acid profile (which seems unlikely); 2) does not have anti-inflammatory properties; or 3) must be administered over a longer time period and/or at a larger dose with greater attention to study participants' fatty acid profiles and whether they change over the course of the study period. It will be challenging to navigate this core issue.

Aspects of the text could be much more succinct, e.g. in the introduction, the rationale/justification for the study can be addressed without including the nuances of supplement research/clinical trials and the retraction of the previous study. Instead, focusing on why n-7 were a compelling target for intervention and expected to influence the outcomes of interest would be sufficient.

The fatty acid composition analysis is not well-described—was this the plasma, cholesterol ester, all fatty acid fractions combined?

Minor

The authors state that participants were fairly health conscious; however, based on their BMIs, one must consider the following possibility:

Poor dietary/exercise habits à Increasing BMI / Obesity à Improved diet/exercise habits

With that in mind and as the investigators acknowledge, it is well established that greater circulating inflammatory marker levels are typical in individuals with excess adipose tissue/visceral fat stores. The possibility that a 3-week fatty acid intervention would be adequate to suppress inflammation and inflammatory pathways in those with these metabolically active fat stores does not seem likely. And while the double cross over design addresses the confounding that adipose tissue and obesity may impose, it may also skew the results toward the null if most of the participants are overweight or obese. A descriptive analysis of participants in whom circulating inflammatory markers were lowered with the intervention may help in examining this possibility.

The statistical methods appear to be appropriate. However, the authors should consider a Bonferroni-corrected significance threshold for the subjective outcomes. It would appear that a priori hypotheses were justified for plasma cytokine levels based on previous evidence; however, evidence that the subjective outcomes would be affected by n7 intervention over a 3-week period was not clear.

The CIH abbreviation was not defined

The figure 3 y-axis label does not make sense for the subjective measures of well-being/QOL

The discussion specifies that there are trends in the data, but the results of statistical trend tests are not disclosed.

Round 2

Reviewer 1 Report

no further comments

Author Response

Minor Revisions:

Reviewer 1: No additional comments or requests for revision.

We further refined the manuscript based on other input. Thank you.

Reviewer 2 Report

Minor comments:

1) Methods for measuring cytokines are mentioned in the first paragraph of the results. These should be moved to the methods and expanded upon, i.e. by what analytical method and on what platform were they quantified.

2)  There are citations and elements of discussion in the results section--paragraphs 2 (page 4 of 12) and the last paragraph of the results. This section should be confined to the results of this trial.

3) Section 3.4 of the results states that free fatty acids were measured; however, the methods section states that the total plasma fatty acids were measured. These are two different measurements. Total fatty acids are composed of the free fatty acids as well as the phospholipid, cholesterol ester, and triglyceride fatty acid fractions, while free fatty acids are a subset of total fatty acids.

4) If the plasma total fatty acid profiles were measured, the corresponding limitation should be re-assessed.

5) Future investigators should also consider conducting a proof-of-principle intervention to demonstrate omega-7 supplementation results in greater plasma or membrane levels and also tests the other fatty acid metabolites of the n7 FAs.

Author Response

Reviewer 2:

Minor comments:

  • Methods for measuring cytokines are mentioned in the first paragraph of the results. These should be moved to the methods and expanded upon, i.e. by what analytical method and on what platform were they quantified.

Response from Authors: The name of one of the Methods subsections was revised to “Biomolecule Measurement (Plasma Total Fatty Acids/Cytokines/hsCRP).” The analytical laboratories are now listed in this section and a description of the cytokine measurement analytical method and on what platform is included.

  • There are citations and elements of discussion in the results section--paragraphs 2 (page 4 of 12) and the last paragraph of the results. This section should be confined to the results of this trial.

Response from Authors: Thank you for this important technical point. We have moved the citations and elements of discussion to the Discussion section and the last paragraph of the Results section was moved to the Discussion section.

  • Section 3.4 of the results states that free fatty acids were measured; however, the methods section states that the total plasma fatty acids were measured. These are two different measurements. Total fatty acids are composed of the free fatty acids as well as the phospholipid, cholesterol ester, and triglyceride fatty acid fractions, while free fatty acids are a subset of total fatty acids.

Response from Authors: “Free” was emphasized for Line 224 because our analysis was not geared to quantify the phospholipid, cholesterol ester or triglyceride form of palmitoleic acid. After chloroform-methanol extraction, the total fatty acids is the accurate description. 

  • If the plasma total fatty acid profiles were measured, the corresponding limitation should be re-assessed.

Response from Authors: Thank you very much for adding this important analytical detail. We include an additional reference regarding the comparison of chloroform-methanol extraction verses solvent-free triglyceride analyses of fatty acids (Bennett, 2007).

  • Future investigators should also consider conducting a proof-of-principle intervention to demonstrate omega-7 supplementation results in greater plasma or membrane levels and also tests the other fatty acid metabolites of the n7 FAs.

Response from Authors: The suggested paragraph was added to the future suggestions.